# Upregulation of Thr/Tyr kinase Increases the Cancer Progression by Neurotensin and Dihydropyrimidinase-Like 3 in Lung Cancer

**DOI:** 10.3390/ijms21051640

**Published:** 2020-02-28

**Authors:** Ying-Ming Tsai, Kuan-Li Wu, Yung-Yun Chang, Jen-Yu Hung, Wei-An Chang, Chao-Yuan Chang, Shu-Fang Jian, Pei-Hsun Tsai, Yung-Chi Huang, Inn-Wen Chong, Ya-Ling Hsu

**Affiliations:** 1Graduate Institute of Medicine, College of Medicine, Kaohsiung Medical University, Kaohsiung 807, Taiwan; yingming@kmu.edu.tw (Y.-M.T.); 980448kmuh@gmail.com (K.-L.W.); cyy807@gmail.com (Y.-Y.C.); jyhung@kmu.edu.tw (J.-Y.H.); chaoyuah@kmu.edu.tw (C.-Y.C.); chienfang1216@gmail.com (S.-F.J.); kanginbobo@gmail.com (P.-H.T.); beryl1992@gmail.com (Y.-C.H.); chong@kmu.edu.tw (I.-W.C.); 2Division of Pulmonary and Critical Care Medicine, Kaohsiung Medical University Hospital, Kaohsiung 807, Taiwan; 960215kmuh@gmail.com; 3School of Medicine, College of Medicine, Kaohsiung Medical University, Kaohsiung 807, Taiwan; 4Division of General Medicine, Kaohsiung Medical University Hospital, Kaohsiung 807, Taiwan; 5Graduate Institute of Clinical Medicine, College of Medicine, Kaohsiung Medical University, Kaohsiung 807, Taiwan; 6Department of Anatomy, Kaohsiung Medical University, Kaohsiung 807, Taiwan; 7Department of Medical Research, Kaohsiung Medical University Hospital, Kaohsiung 807, Taiwan; 8Drug Development and Value Creation Research Center, Kaohsiung Medical University, Kaohsiung 807, Taiwan; 9Graduate Institute of Medicine, College of Medicine, Kaohsiung Medical University, No. 100, Shih-Chuan 1st Road, Kaohsiung 807, Taiwan

**Keywords:** dihydropyrimidinase-like 3, lung cancer, neurotensin, spindle assembly checkpoint, threonine tyrosine kinase

## Abstract

Lung cancer is one of the leading causes of cancer-related death globally, thus elucidation of its molecular pathology is highly highlighted. Aberrant alterations of the spindle assembly checkpoint (SAC) are implicated in the development of cancer due to abnormal cell division. TTK (Thr/Tyr kinase), a dual serine/threonine kinase, is considered to act as a cancer promoter by controlling SAC. However, the mechanistic details of how TTK-mediated signaling network supports cancer development is still a mystery. Here, we found that TTK was upregulated in the tumor tissue of patients with lung cancer, and enhanced tumor growth and metastasis in vitro and in vivo. Mechanistically, TTK exerted a significant enhancement in cancer growth by neurotensin (NTS) upregulation, and subsequently increased the expression of cyclin A and cdk2, which was resulting in the increase of DNA synthesis. In contrast, TTK increased cell migration and epithelial-to-mesenchymal transition (EMT) by enhancing the expression of dihydropyrimidinase-like 3 (DPYSL3) followed by the increase of snail-regulated EMT, thus reinforce metastatic potential and ultimately tumor metastasis. TTK and DPYSL3 upregulation was positively correlated with a poor clinical outcome in patients with lung cancer. Together, our findings revealed a novel mechanism underlying the oncogenic potential effect of TTK and clarified its downstream factors NTS and DPYSL3 might represent a novel, promising candidate oncogenes with potential therapeutic vulnerabilities in lung cancer.

## 1. Introduction

Lung cancer is the leading cause of cancer death worldwide [1,2]. Although tremendous efforts have been made to investigate the tumorigenesis and cancer development, the underlying mechanisms still remain elusive. Recently, technological advances such as microarrays and next-generation sequencing (NGS) have permitted for the in-depth genomic depiction of variety tumors, which increase our thoughtfulness of cancer initiation and advancement [3,4]. Numerous genetic and epigenetic variations relevant to lung development are established through these techniques. However, understanding the function and its regulatory network of individual oncogenes are insufficient, despite the accumulating massive data.

Dual serine/threonine kinase TTK (Thr/Tyr kinase), also known as monopolar spindle 1 (Mps1), plays an important role in spindle assembly checkpoint (SAC) by regulating association of the mitotic checkpoint complex to unattached kinetochores [5,6]. TTK has been considered to be dysregulated in various cancer cells because TTK dysregulation causes excess centrosomes resulting in aberrant mitotic spindles. Reduced in TTK level or activity in tumors can lead to a decrease of cell viability and division; thus, inhibition of TTK has been regard as an attractive target for anti-cancer drug development [6,7,8]. TTK has been also reported that upregulation of TTK increases lung cancer progression due to X-linked deubiquitinase USP9X dysfunction [9,10]. With several therapeutic agents targeting TTK currently being evaluated as chemotherapy agents in clinical trials, better understanding of the molecular mechanisms mediating TTK-mediated oncogenic cascade could have a noteworthy impact by directorial their successful clinical application and preventing chemo-resistance development.

Our previous study showed that elevated levels of TTK were found in the tumor parts of lung cancer using NGS [11]. In this study, we uncover a previously unknown mechanism by which TTK exerts a significant oncogenic effect by the upregulation of neurotensin (NTS) and dihydropyrimidinase-like 3 (DPYSL3), which promote cancer growth and cancer progression consequently tumor metastasis. The objective of the study herein illustrates the functional consequences and molecular mechanism of the TTK expression in lung cancer and evaluates its potential significance regarding the beneficial vulnerabilities of lung cancer therapy.

## 2. Results

### 2.1. Elevated TTK Expression Conferred Poor Prognosis in Lung Cancer Patients

TTK is tightly regulated during cancer development [12], however, the molecular mechanisms of TTK remains unknown. To prove the role of TTK involving in lung cancer, we assessed the expressions of TTK from normal and cancer parts of two lung cancer patients. In these two matched pairs of human specimens, high levels of TTK have been found in cancer parts comparing with normal parts at mRNA level as determined by NGS (Figure 1A). In addition, the five cohorts form Oncomine® database showed that levels of TTK expression were higher in cancer parts than normal parts in lung cancer patients (Figure 1B). GSE31210 dataset [13] also further confirmed that a higher level of TTK expression in lung cancer parts than normal parts, although it did not show a significantly difference between patients with stage I or stage II (Figure 1C). To validate it, immunohistochemical (IHC) staining of TTK showed the higher expressions in lung cancer parts when comparing with normal parts (Figure 1D) from eight (six in Figure 1D and two in Figure 6E) out of nine patients with lung cancer. To further validate TTK’s role in lung cancer prognosis, the overall survival analyses of GEPIA [14], Kaplan–Meier’s Survival Analysis and the Okayama (GSE31210) cohort in the Kaplan-Meier’s (KM) plotter [15] revealed patients of higher levels of TTK expression survived shorter than in lung cancer patients with lower expression levels (Figure 1E). To conclude these results, TTK were expressed in lung cancer part and higher levels of TTK expression conferred shorter survival time. Taking all three datasets, we speculated that TTK might play an oncogenic role in lung cancer development.

### 2.2. TTK Decreased Cancer Growth by Interfering Cell Cycle Progression in Lung Cancer 

TTK is a serine/threonine kinase that controls the cell proliferation through mitosis by modulating the SAC [16], we thus silenced TTK expression by shRNA plasmid transfection in lung cancer for investigating the role of TTK in lung cancer growth. As shown in Figure 2A, the TTK-knockdown stable A549 clone was successfully established. The long-term and short-term proliferation of A549 was analyzed through several methods. The long-term cancer growth carried out by colony formation revealed TTK knocking down reduced lung cancer cell proliferation (Figure 2B). WST-1 analysis further confirmed TTK knockdown suppressed lung cancer cells proliferation after 3 days’ culture (Figure 2C). In addition, BrdU incorporation showed that suppression of TTK expression led to decreased A549 cell proliferation by inhibiting DNA synthesis (Figure 2D). To investigate the molecular mechanism of cell proliferation inhibited by TTK knockdown, cell cycle-related proteins were investigated. The results showed that the S phase related protein, such as Cyclin A and Cdk2 were decreased in TTK-knockdown stable A549 clone (Figure 2E). Consistently, the cell cycle analysis revealed the S-phase population was decreased (from 19.45% to 12.37%) in TTK knockdown cancer cells which is consistent with BrdU incorporation analysis (Figure 2F). 

### 2.3. TTK Regulated Metastatic Behaviors and EMT in Lung Cancer 

As shown in Figure 3A, the migratory ability of A549 and CL1-5 cells was suppressed after TTK knockdown via wound healing assay (Figure 3A). The transwell assay revealed attenuated cell migration more than 50% after TTK knockdown in both A549 and CL1-5 cell lines (Figure 3B). 

Epithelial–mesenchymal transition (EMT) is involved in cancer metastasis and treatment resistance [17]. The EMT markers revealed higher expression levels of epithelial characters such as E-cadherin but lower expression levels of mesenchymal characters such as N-cadherin, vimentin, α-SMA and Snail in TTK knockdown-A549 and CL1-5 cells (Figure 3C). In addition, with the TTK knockdown in both cell lines, cell invasive ability declined as determined by an ECM-coated invasion assay (Figure 3D). These results meant TTK knockdown suppressed cancer metastatic characters as migration, invasive and EMT. 

### 2.4. NTS and DYPSL3 Act as Downstream Regulators of TTK

To study downstream signaling pathways regulated by TTK, the gene profiles of TTK knockdown A549 cells were assayed by microarray. The heat map of gene prolife revealed a different expression pattern after the TTK knockdown (control plasmid transfected cells versus TTK shRNA plasmid transfected cells), which showed 8 genes upregulation and 11 genes downregulation. The most significant downregulated genes after TTK knockdown in A459 cells were HOXD-AS2, ITM2A, NTS, DPYSL3 and SPRAC and they were expressed below 0.5-fold change (Figure 4A). We validated microarray data by qRT-PCR showed that only two genes, NTS and DPYSL3, were decreased in A549 cells after TTK knockdown (Figure 4B). Moreover, the levels of both NTS and DPYSL3 proteins were reduced in TTK-silencing A549 cells (Figure 4C,D), suggesting NTS and DPYSL3 act as downstream molecules TTK. 

### 2.5. NTS Is Involved in TTK- Related Cell Proliferation Regulation

To study whether NTS contributes TTK-mediated cancer growth and progression, we used exogenous NTS to reveal the change of cancer proliferation and migration augmented by TTK. NTS of concentration 100 nM was added to study. The inhibitory effect of TTK knockdown in the cell proliferation of A549 cells was reversed by exogenous NTS (100 nM), while NTS did not affect the proliferation of A549 cells at the same concentration (Figure 5A). On the contrary, the migratory inhibition mediated TTK knockdown was not reversed by exogenous NTS in A549 cells, as determined by both wound-healing (Figure 5B) and transwell analysis (Figure 5C). Consistently, the inhibitory effect of TTK knockdown in the expressions of cell cycle related proteins, Cyclin A and Cdk2, were aborted by exogenous NTS (Figure 5D). The oncogenic role of NTS was supported by the KM plotter analysis, which showed the higher expression group of NTS survived for a shorter time than the lower expression group. These results suggested that NTS reversed the cell division caused by TTK knockdown but not migration ability of lung cancer cells. Furthermore, the high-expression of NTS conferred poor prognosis of survival in lung cancer patients.

### 2.6. DPYSL3 Contributes the Enhancement of Cell Migration and EMT

To determine the role of DPYSL3 in the migration and EMT, DPYSL3 was inhibited using siRNA transfection in A549 cells. The knockdown efficiency of DPYSL3 by siRNA transfection reached 80%, compared to control siRNA transfected cells (Figure 6A). Knockdown of DPYSL3 by siRNA decreased cell migration and invasion in A549 cells (Figure 6B,C). In addition, the knockdown of DPYSL3 increased epithelial markers such as E-cadherin whereas it decreased mesenchymal markers such as N-cadherin, vimentin and α-SMA (Figure 6D). The correlation of DPYSL3 and TTK expression was also observed in the sequential tissue section of tumor part of lung tissue obtained from patients with lung cancer (Figure 6E). It implied TTK regulated the expression of DPYSL3. Moreover, the DPYSL3 levels in the tumor of lung cancer patients were negatively correlated with the overall survival in two databases (ProgGeneV2, KM plotter and GSE31210; Figure 6F). In short, higher DPYSL3 levels were associated with poor clinical outcome due to the effects of DPYSL3 on the regulation of cancer migration, invasion and EMT.

### 2.7. Inhibition of TTK Suppressed Cancer Growth and Lung Metastasis In Vivo

To verify the oncogenic role of TTK in cancer growth in vivo, control and TTK-knockdown A549 were implanted into the flank of nude mice. As shown in Figure 7A, the tumor volume of TTK-knockdown A549 were smaller than the tumor of control A549 after 52-days observation (Figure 7A). The expression levels of TTK in tumors revealed successful knockdown of TTK by shRNA in vivo (Figure 7B).

Next, we assessed the role of TTK on lung cancer metastasis, control and TTK-knockdown A549 were injected into mice via the tail vein as a metastasis model. Figure 7C,D shows that the number of tumor nodule in the lungs of TTK-knockdown A549-beraing mice was much less than that in the lungs of control A549 mice. IHC staining reveals that the expression of TTK in the tumor sections of control A549 mice was higher the tumor sections of lungs of TTK-knockdown A549-beraing mice (Figure 7E). These results suggested that inhibition of TTK decreased cancer growth and metastasis in vivo.

## 3. Discussion

Using both NGS and bioinformatics approaches to characterize the gene profile of lung cancer, we identified TTK had the oncogenic potential in lung cancer, compared to non-cancerous cells. TTK is regulated in multiple types of cancers, including breast, liver, lung and pancreatic cancer [6,18,19]. Despite its basic biological function and clinical significance, the underlying mechanisms of TTK-mediated cancer progression remain poorly understood. In this study, we further demonstrated that TTK enhances cell proliferation, migration, EMT and invasion in lung cancer. The expression of TTK in tumor regions is higher than that in non-tumor regions. Online dataset also indicated the TTK expression was associated with shorter overall survival rate. Therefore, the expression of TTK in tumor tissue may serve as a biomarker for predicting metastatic risk in lung cancer (Figure 8). 

Neurotensin (NTS) is an endogenous neuropeptide and widely expressed in the central nervous system [20]. NTS has been identified as a local hormone and plays a critical role in the regulation of fat storage, obesity and metabolic disorders. Previous studies also revealed an association of NTS with progression and invasiveness of various cancers, thus is considered to be a potential pharmacological target in cancer therapy [21,22,23,24]. NTS increases cell proliferation by a protein kinase C (PKC)-dependent ligand-induced transactivation of EGFR, which in turn activates the Raf-mitogen-activated protein kinase (MEK)-ERK signaling transduction [25,26,27]. High expression of NTS and its receptor NTSR1 are identified in lung cancer, compared with normal cells [28]. In this study, we found that inhibition of TTK increased the expression of NTS, which was observed by a microarray, but also are validated by the protein expression analysis. Furthermore, addition of exogenous NTS restored cell proliferation, cyclin A and cdk2 expression, which were inhibited by TTK deficiency. Higher level of NTS in lung cancer patients was positively associated with a poor outcome, representing that NTS contributes the malignant development of lung cancer. Therefore, this current study verifies that NTS plays a critical role in TTK-mediated breast cell proliferation, and if targeted, may have the potential therapeutic benefit to reduce lung progression. 

DPYSL3, also referred to as the collapsing response mediator protein 4, is a member of the DPYSL gene family that is highly expressed in both normal tissues and tumors derived from the lung, colon and prostate [29]. The functions of DPYSL3 involves in various cellular processes, including cell differentiation, migration, neurogenesis and neurodegeneration [30]. Conflicting roles of DPYSL3 have been described in different types of cancer, implying that it has a diversity of functions in different malignancies. Previous studies indicated that DPYSL3 inhibits cell migration by regulating the actin cytoskeleton in neuroblastoma cells [29]. DPYSL3 is also reported to play a role in cell migration and metastasis suppression in liver and prostate cancer [31,32,33]. In contrast, DPYSL3 is positively associated with metastasis and a poor outcome in breast cancer, renal cell carcinoma, gastric and pancreatic cancer [34,35,36,37]. Our study indicates that DPYSL3 acts as a promoter of cancer progression supported by the loss of DPYSL3 reduced cancer migration and EMT. Consequently, a high expression of DPYSL3 in the patients was strongly associated with shortened survival. Although further exploration is required to clarify the underlying molecular mechanism how TTK regulates DPYSL3 expression and then potentiates malignant behavior, our study offers a valuable insight for the specific management patients with lung cancer.

Taken together, our study offers insights into TTK as an oncogene that regulates cell proliferation, movement and EMT in lung cancer. These are important developments in malignant cells, and TTK expression could thus be used to stratify lung patients for future clinical trials that inhibit TTK and its targets.

## 4. Materials and Methods

### 4.1. Cell Lines

Human lung adenocarcinoma cell line A549 cells were obtained from the American Type Culture Collection (ATCC, Manassas, VA). CL1–5 cells were kindly provided by Dr. Pan-Chyr Yang of National Taiwan University. A549 was cultured in F-12K Medium (ATCC) supplemented with 10% fetal bovine serum (FBS), 100 U/mL penicillin and 100 μg/mL streptomycin (Thermo Fisher Scientific, Boston, MA). CL1-5 cells were cultured in RPMI1640 medium (Lonza, Basel, Switzerland) supplemented with 10% FBS and antibiotics, as described above. NTS (Neurotensin) (Sigma-Aldrich, St. Louis, MO, USA) were used at a working concentration of 100 nM. A549 cells were authenticated by a short tandem repeat (Promega, Madison, WI) and examined negative for mycoplasma contamination by MycoAlert™ mycoplasma detection kit (Lonza, Switzerland) every 3 months.

### 4.2. Bioinformatics

The level of mRNA expression in lung cancer and normal specimens (cancer vs. normal) were extracted from the Oncomine database (http://www.oncomine.org, Compendia biosciences, Ann Arbor, MI, USA). The criteria in the analysis were fold change > 2 and *p*-value < 0.05, which was calculated using the Oncomine database through a two-sided Student’s *t*-test. The correlation of specific genes and overall survival rates in lung cancer was assessed by the KM plotter (http://kmplot.com/analysis/) and GEPIA (http://gepia.cancer-pku.cn/). Patients were divided into 2 groups with the best cut-off, which was computed with median survival. The hazard ratios (95% confidence intervals) were calculated using the Cox proportional model. The microarray database (GSE31210 and GSE30129) of normal and lung cancer tissues was also collected from the Gene Expression Omnibus (GEO) database. 

### 4.3. NGS and Microarray

The deep sequencing for the pairs of lung adenocarcinomas and adjacent normal tissue and microarray for TTK knockdown A549 cells were performed by a biotechnology company (Welgene, Taipei, Taiwan). Correspondingly fragmented Cy3 (CyDye, Agilent Technologies, Santa Clara, CA, USA) labeled cRNA was pooled and hybridized to Agilent SurePrint Microarray (Agilent Technologies, Santa Clara, CA, USA) and then scanned with an Agilent microarray scanner. Data was analyzed by Feature extraction 10.7.3.1 software and raw signal data was normalized by quantile normalization for differential expressed genes discovering. The criteria for differentially expressed mRNA by NGS analysis were fold change > 2 and fragments per kilobase million (FPKM) > 0.3. 

### 4.4. RNA-Sequencing and Quantitative Real-Time Polymerase Chain Reaction (Qrt-PCR)

The pairs of adjacent non-tumor lung and tumors were harvested from the Division of Thoracic surgery and Division of Pulmonary and Critical Care Medicine, Kaohsiung Medical University Hospital (Kaohsiung, Taiwan). Total RNA was isolated from cells using the TRIzol Reagent (Life Technologies, Carlsbad, CA, USA) and cDNA were reverse transcribed using reverse transcriptase kits, respectively (Takara, Shiga, Japan). RNA levels were determined using real-time analysis with SYBR Green on a StepOne-Plus machine (Applied Biosystems, Foster City, CA, USA). The relative expression levels of the specific mRNA were normalized to glyceraldehyde 3-phosphate dehydrogenase. The relative standard method (2^−ΔΔCt^) was used to calculate relative RNA expression. The following primers were used: TTK (forward, 5′- CACCACAAGATGCAGAAATAGG-3′ and reverse, 5′-CCAAATCTCGGCATTCTGAT-3′); HOXD-AS2 (forward, 5′- AGCAGCAACTTGACCCAAAC -3′ and reverse, 5′-TTGAGAATCCTGGCCACCTC -3′), NTS (forward, 5′- TATGCATGCTACTCCTGGCT -3′ and reverse, 5′- CTGTTTCCTCAGCTGGGC -3′), DPYSL3 (forward, 5′- CACAAACGCTGCCAAGATCT -3′ and reverse, 5′- AGATGACAACCAGAGGAGCC -3′), ITM2A (forward, 5′- AGAGCCTAACTTCCTGCCTG -3′ and reverse, 5′- CAGTTCCCCAGCAACAAGTC -3′), SPARC (forward, 5′- GTGCAGAGGAAACCGAAGAG-3′ and reverse, 5′- AGTGGCAGGAAGAGTCGAAG -3′) and GAPDH (glyceraldehyde 3-phosphate dehydrogenase) (forward, 5′-TTCACCACCATGGAGAAGGC-3′ and reverse, 5′-GGCATGGACTGTGGTCATGA-3′).

### 4.5. Immunoblot and NTS Determination

The total cellular protein was extracted using the RIPA lysis buffer (EMD Millipore, Billerica, MA, USA) supplemented by a protease inhibitor cocktail (Sigma-Aldrich, St. Louis, MO, USA). An equal amount of cellular protein was denatured by heating, and then separated by SDS-PAGE. Proteins in the gel were transferred to polyvinylidene difluoride membranes (EMD Millipore), which were probed with various primary antibodies for 4–16 h after blocking in TBST containing 5% milk, followed by incubation with horseradish peroxidase (HRP)-conjugated secondary antibodies (Cell-Signaling Technology, Danvers, MA). The signal of the specific protein was detected using a chemiluminescence kit (EMD Millipore). Primary antibodies used include those against TTK (Catalog #3255), Slug (Catalog #9585), Snail (Catalog #3879), Cyclin A (Catalog #4656), Cyclin B (Catalog #4138), Cyclin D (Catalog #2978), Cyclin E1 (Catalog #4129), Cyclin E2 (Catalog #4132), cdk2 (Catalog #2546), cdk4 (Catalog #2906) and cdk6 (Catalog #3136) were obtained from Cell Signaling Technology (Carlsbad, USA). Anti-N-Cadherin (Cat.610921), E-cadherin (Cat.610182) and Vimentin (Cat.550513) antibodies were pursed from Becton Dickinson biosciences. Anti-Actin, α-Smooth Muscle antibody (A5228,) and GAPDH (catalog# MAB374) antibodies pursed from EMD Millipore. Anti-DPYSL3 Polyclonal Antibody was obtained from Thermo Fisher Scientific, Inc., (Wilmington, DE, USA). The quantitation result of the immunoblot was performed by AlphaImager software (Alpha Innotech, San Leandro, CA, USA). The supernatants of A549 cell transected with control shRNA or TTK shRNA plasmid were harvested after 48 h culture. The levels of NTS were measured by Human Neurotensin (NTS) ELISA Kit (Cat. CSB-E09144h, CUSABIO, Houston, TX).

### 4.6. TTK and DPYSL3 Knockdown

Knockdown of TTK in A549 cells was performed using an shRNA expression system obtained from the National RNAi Core Facility (Taipei, Taiwan). The stable clone of TTK knockdown cells were established by the puromycin section. A549 cells were transfected with the control or DPYSL3 ON-TARGET plus SMARTpool siRNA using Dharmafect reagents No1 (Dharmacon, Lafayette, CO, USA). The knockdown efficacy of TTK shRNA plasmid or DPYSL3 siRNA determined by an Immunoblot and qRT-PCR, respectively. 

### 4.7. Cell Proliferation, Colony Formation and 5-Bromo-2-Deoxyuridine (Brdu) Incorporation 

Cells (3 × 10^3^ cells/well) were seeded in a 96 well plate, then cultured for 3–5 days, the cell proliferation was assessed by cell proliferation reagent WST-1 according to the manufacturer’s instructions. Cells were labeled with BrdU (10 μM) at day 2 after seeding followed by fixation. Incorporated BrdU was detected by the ELISA-based method according to the manufacturer’s protocol (BrdU Cell Proliferation Assay Kit, (EMD Millipore)). For the colony formation analysis, cells were inoculated in 6-well plates at the density of 2000 cells/plate. Culture medium was changed every three days. Cells were allowed to grow for 7 days to form colonies, which was fixed by 4% paraformaldehyde and stained by crystal violet. The number of colonies containing ≥ 50 cells was calculated under a microscope.

### 4.8. Cell Migration and Invasion 

Cells were seeded in to a 12 well-pate at 90% confluence, and the cell migration was determined by measuring the movement of cells into the acellular area created by a sterile tip. The wound closure was observed after 24 h. Quantitative determination of cell migration was also performed by the transwell system. Cells (1.5 × 10^5^ cells/insert) were seeded to the transwell and complete containing 10% FBS was filled to bottom well. After 24 h, cells on the upper side of the membrane surface were removed by scraping with a cotton swab, the migratory cells were fixed in 4% paraformaldehyde for 20 min and stained with 0.1% crystal violet for overnight. The invading cells were quantified by invasion assay kits (Millipore, Bedford, MA) using fluorescence dye staining, and the optic density (OD) value read by a fluorescence plate reader at excitation/emission wavelengths of 485/530 nm. 

### 4.9. Cell Cycle Analysis

Control shRNA and TTK shRNA transfected A549 cells were collected by trypsinization, washed with PBS and fixed with cold 100% ethanol. Cells were stained by Propidium Iodide (20 μg/mL; Life Technologies) in PBS buffer containing 0.1% triton-X-100and RNase A (100 μg/mL; Sigma) for 30 min. The suspension was passed through a nylon mesh filter and analyzed using a BD Accuri™ C6 flow cytometry.

### 4.10. Animal Models 

Control plasmid transfected and TTK-knockdown A549 (2 × 10^6^) were injected subcutaneously into the flanks of nude mice (male, 6-week old, *n* = 6–8). The tumor volumes of the mice were measured every 3 days. Tumor volume was assessed by measuring the length (L) and width (W) of the tumor with calipers (tumor volume (mm^3^) = 0.5 × L × W^2^). For metastatic analysis, control plasmid transfected and TTK-knockdown A549 were transplanted into nude mice by tail vein injection. Animals were sacrificed on weeks 12 and the number of tumor nodules in lung was counted. All mice were obtained from the Vital River Laboratories (Beijing, China) and housed in a specific pathogen-free environment. Two experiments were conducted in accordance with the National Institutes of Health Guide for the Care and Use of Laboratory. The animals used were approved by the KMU Animal Care and Use Committee. 

### 4.11. Immunohistochemistry (IHC)

The tissue was obtained from 10 lung cancer patients admitted to the Division of Pulmonary and Critical Care Medicine at Kaohsiung Medical University Hospital (KMUH), Kaohsiung, Taiwan. The Institutional Review Board of KMUH approved the study’s protocol, and all participants provided written informed consent in accordance with the Declaration of Helsinki. The expressions of TTK and DPYSL3 were demonstrated using anti-TTK (dilution 1:200, GTX16407) and anti-DPYSL3 (dilution 1:100, GTX16407, GeneTex Ltd. Irvine, CA, USA) antibodies, respectively. All of the sectioned tissues were counterstained with hematoxylin. 

### 4.12. Statistical Analysis

Results are presented as mean ± standard deviation (SD). Multiple group comparisons were calculated by a two-way analysis of variance (two-way ANOVA) with a Tukey’s post-hoc test used with the assistance of the GraphPad Prism program (7.04 version, Graphpad Software, San Diego, CA). Two tested groups were compared using a Student’s *t*-test. Results were considered statistically significant when *p* value < 0.05.

## Figures and Tables

**Figure 1 ijms-21-01640-f001:**
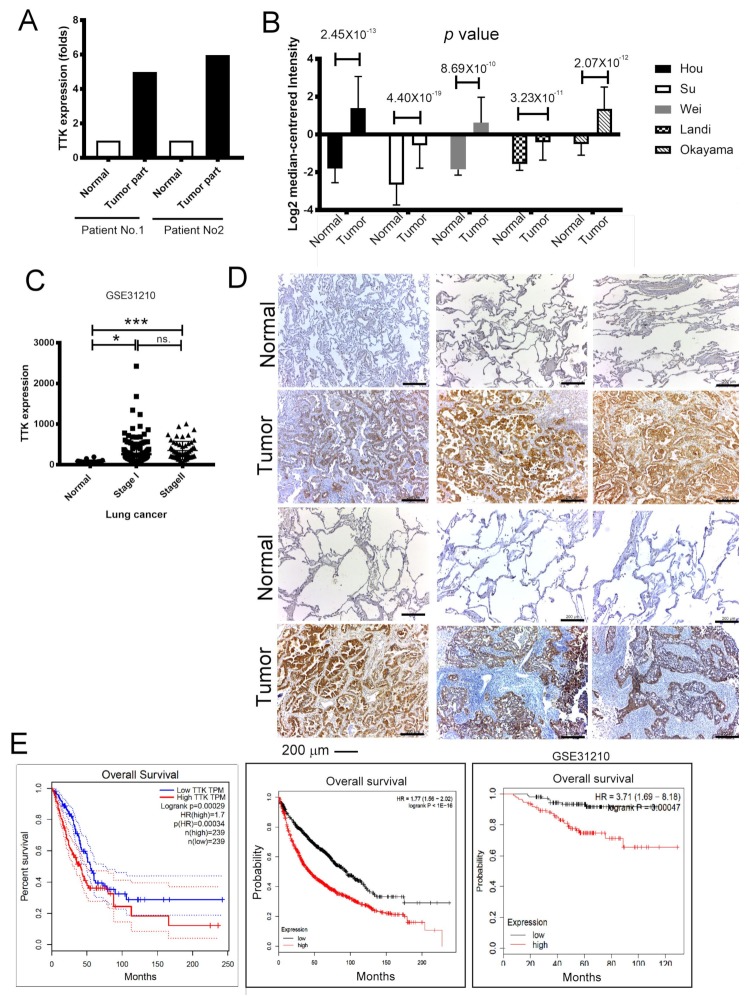
Upregulation of Thr/Tyr kinase (TTK) is correlated with poor prognosis in lung cancer. (**A**) The levels of TTK in 2 lung cancer patients. Increase TTK is found in different lung cancer patient database obtained from Oncomine® (**B**) and GEO (GSE31210) (**C**). (**D**) The immunohistochemical (IHC) staining of TTK of the specimens of 6 out of 9 patients lung cancer patients, the higher intensity of TTK staining in the tumor parts when comparing with normal lung parts. (**E**) The relation of TTK expression with clinical outcome in lung cancer patients. The group was divided according to GEPIA, the KM plotter website and GSE31210 dataset. * Significant difference between the two test groups (* *p* < 0.05, *** *p* < 0.005).

**Figure 2 ijms-21-01640-f002:**
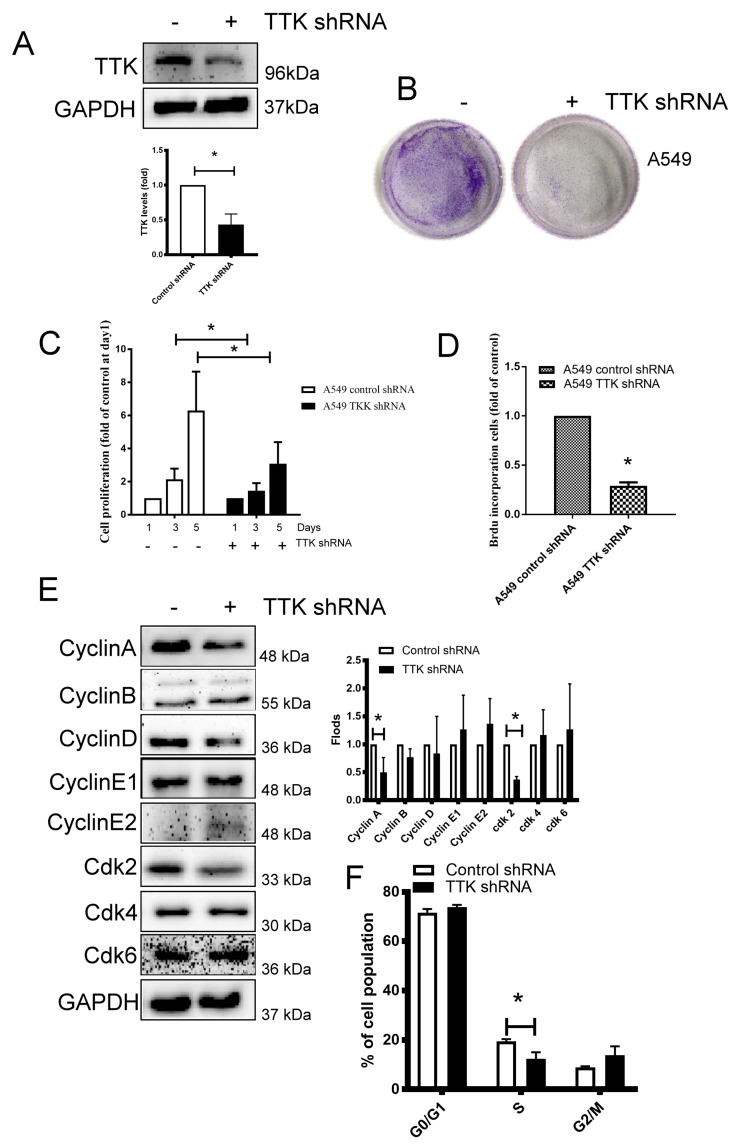
Knockdown of TTK inhibits cancer proliferation. (**A**) The efficacy of TTK knockdown in A549 cells. A549 cells were transfected with control or TTK shRNA plasmid, and the stable clones were established by puromycin selection. The expression of TTK was determined by an Immunoblot. Inhibition of TTK decreased cell proliferation, as determined by colony formation (**B**), WST-1 (**C**) and BrdU incorporation (**D**). The cell proliferation of TTK knockdown A549 cells was determined after 3-5 days of incubation. The colony formation was counted after 7 days of growth. The effect of TTK in cell cycle-related proteins and their quantification (**E**) and cell cycle distribution (**F**). The expressions of various proteins were assessed by an Immunoblot. The cell cycle was determined using a flow cytometry after Propidium Iodide staining. All experiments were performed independently at least three times. * Significant difference between the two test groups (*p* < 0.05).

**Figure 3 ijms-21-01640-f003:**
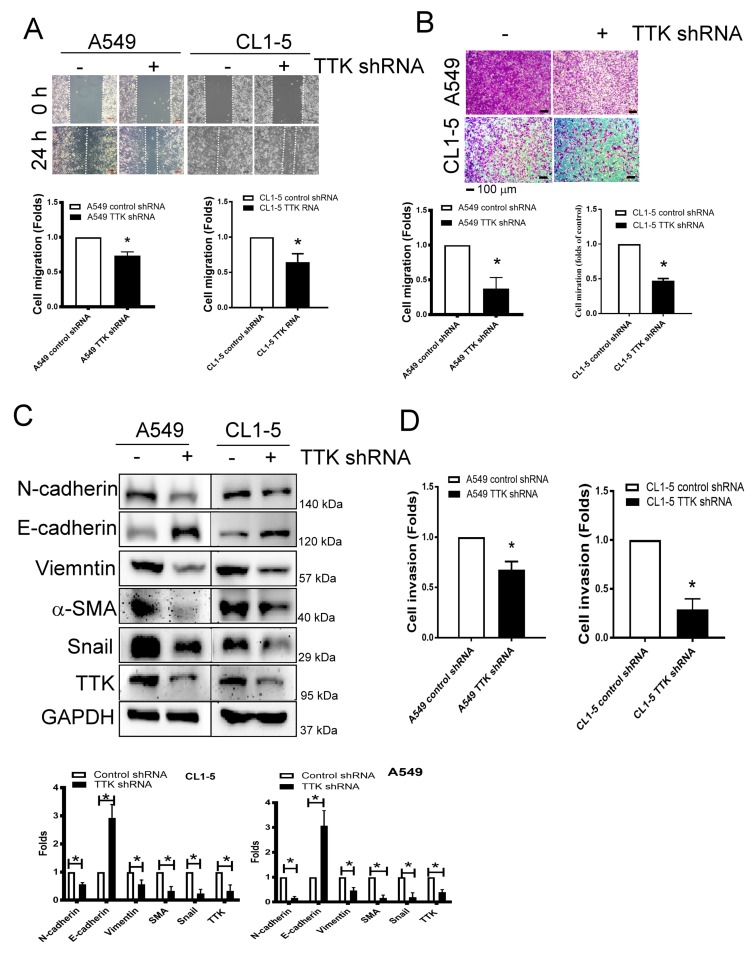
Inhibition of TTK decreases cancer migration and epithelial–mesenchymal transition (EMT). Inhibition of TTK decreased cell migration, as determined by wound healing (**A**) and the transwell system (**B**). Decrease TTK reduced EMT (**C**) and invasion (**D**). Control shRNA or TTK shRNA plasmid transfected A549 and CL1-5 cells were seeded in the upper insert coated with (for invasion analysis) or without (for migration analysis) Matrigel in serum-free conditions and culture medium (10% FBS) was added into the lower well to act as a chemo-attractant for 24 h (for migration) or 48 h (for invasion). The migratory and invasive cells were quantified by crystal violet or fluorescence dye staining. EMT marker expressions were assessed by an Immunoblot. All experiments were performed independently at least three times. * Significant difference between the control and TTK knockdown cells (*p* < 0.05).

**Figure 4 ijms-21-01640-f004:**
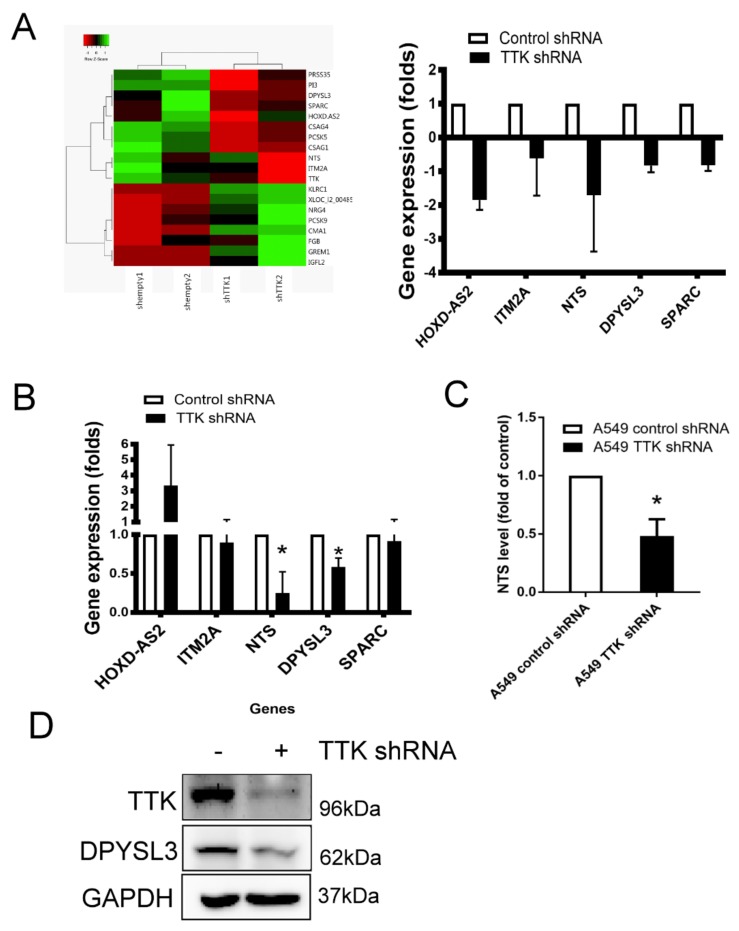
The gene profile of TTK-knockdown A549 cells. (**A**) The heat-map. The gene profile of TTK knockdown A549 cells was established by a microarray. (**B**) The decrease genes of TTK knockdown A549 cells. The expressions of various mRNAs were measured by qRT-PCR. Knockdown of TTK decreased neurotensin (NTS; **C**) and DYPSL3 (**D**) expression protein levels. The level of NTS in the supernatants of control or TTK knockdown A549 cells was determined by ELISA after 48 h incubation. The expression of DYPSL3 was assessed by an Immunoblot. All experiments were performed independently at least three times. * Significant difference when comparing with control group (*p* < 0.05).

**Figure 5 ijms-21-01640-f005:**
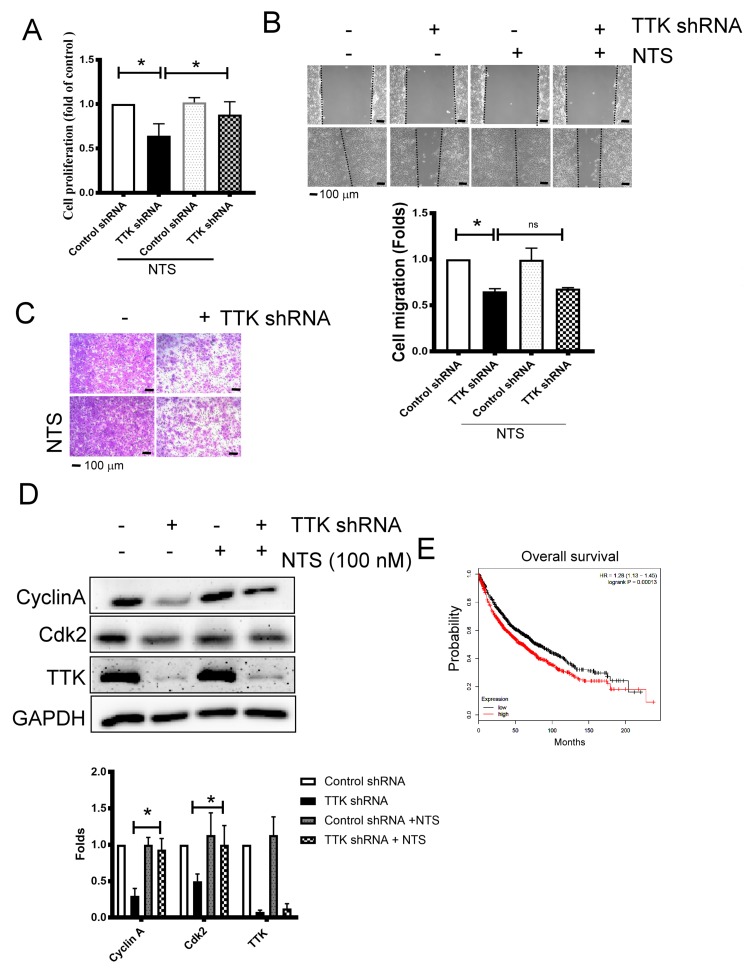
NTS contributes TTK-mediated cell proliferation. (**A**) NTS reversed cell proliferation inhibition (**A**), but not cell migration, as determined by wound healing (**B**) and the transwell system (**C**) by TTK knockdown. The cell proliferation of A549 or TTK knockdown A549 cells was determined by WST-1 after 72 h incubation. Control shRNA or TTK shRNA plasmid transfected A549 were seeded in the upper insert in serum-free medium with or without NTS. The complete culture medium was added into the lower well to act as a chemo-attractant for 24 h. The migratory cells were quantified by crystal violet staining. (**D**) NTS prevented the decrease of cyclin A and cdk2 regulated by TTK inhibition. NTS were added to A549 or TTK knockdown A549 cells for 24 h, the expression of various proteins was determined by an Immunoblot. (**E**) The relation of NTS with overall survival rate of lung cancer patients. The overall survival rate of lung cancer was obtained from the KM plotter website. All experiments were performed independently at least three times. * Significant difference between the two test groups (*p* < 0.05).

**Figure 6 ijms-21-01640-f006:**
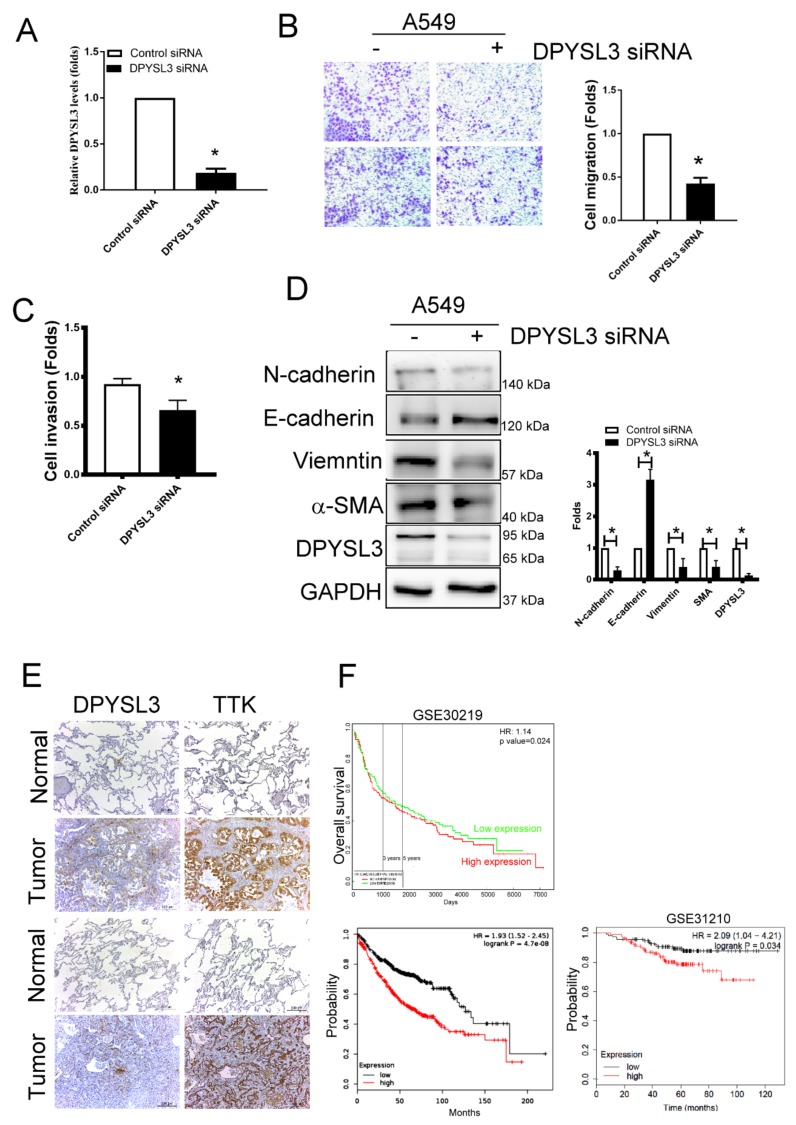
DPYSL3 regulated cell migration and EMT. (**A**) The efficacy of DPYSL3 siRNA transfection. A549 cells were transfected either with control or DPYSL3 siRNA for 24 h. The level of DPYSL3 was determined by qRT-PCR. Inhibition of DPYSL3 decreased cell migration, as determined by the transwell system (**B**). The effect of DPYSL3 on cell invasion was assessed at 24 h post-transfection (**C**). DPYSL3 regulated EMT in lung cancer cells. The expression of various proteins was measured after 48 h of transfection (**D**). The correlation between DPYSL3 and TKK in the sequential lung tissue isolated from patients with lung cancer (**E**). The relation of DPYSL3 with the overall survival rate of lung cancer patients in different cohorts (**F**). All experiments were performed independently at least three times. * Significant difference between the two test groups (*p* < 0.05).

**Figure 7 ijms-21-01640-f007:**
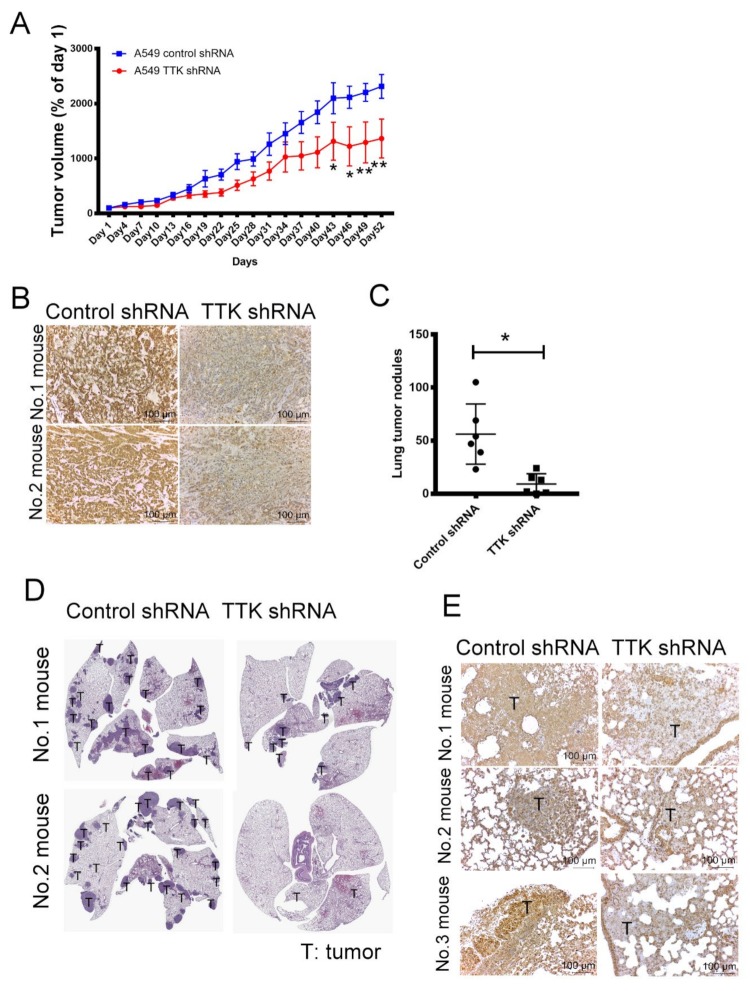
The effect of TTK in cancer growth and metastasis in vivo. (**A**) TTK knockdown decreased tumor growth in nude mice. (**B**) The level of TTK in A549 tumor of mice. Control shRNA and TTK shRNA plasmid transfected A549 cells were administered subcutaneously into the right flank of nude mice. The tumor volume was measured every 3 days. Fifty-three days after the tumor inoculation, tumor mass was excised and the level of TTK was determined by IHC staining. Control shRNA and TTK shRNA-transfected A549 were injected into mice via the tail vein. After 12 weeks, non-tumorous and tumorous regions of the lungs were harvested. (**C**) TTK inhibition reduced lung metastasis in an animal model. (**D**) The H&E staining of tumor nodules in lungs of mice. (**E**) The expression of TTK in lung tumor nodules. Representative tumor sections were stained with TTK antibody and photographed at 200 × magnification. * Significant difference between the two test groups (*p* < 0.05).

**Figure 8 ijms-21-01640-f008:**
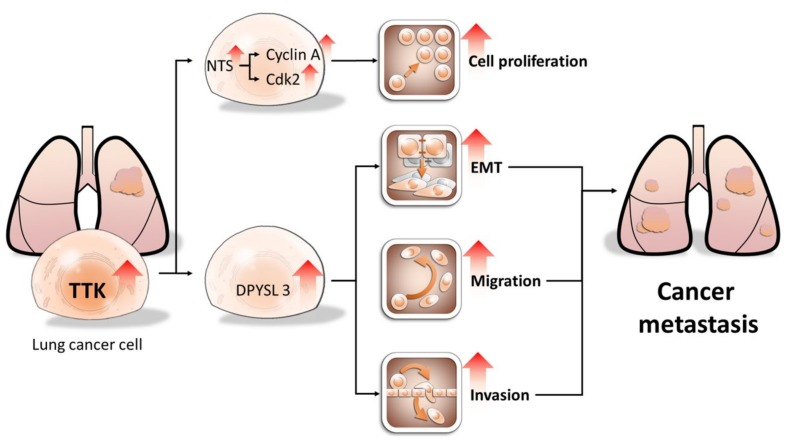
Proposed model of TTK oncogenic potential in lung cancer. TTK increases cell proliferation by enhancing NTS expression, which in turn elevates cyclin A and CDK2 expression. In contrast, TTK promote cancer metastasis by causing EMT in a DPYSL3-dependent manner. Our study may provide novel target for developing personalized diagnostics and therapeutic strategy for lung cancer patients. The arrows mean up-regulation of levels of specific genes and cellular behaviors.

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
