# Peer review of "Upregulation of Thr/Tyr kinase Increases the Cancer Progression by Neurotensin and Dihydropyrimidinase-Like 3 in Lung Cancer"

_ijms, 2020, doi:10.3390/ijms21051640_

Round 1
Reviewer 1 Report
In this article, Dr. Ying-Ming Tsai and colleagues presented new results about involving of serine/threonine kinase TTK in development of lung cancer. The paper is well written and potentially contains new information about TTK signaling. However, there are a major problems without correcting which this article cannot be published. I noticed the following problems:
The Figure 1D is poorly described. In figure legend or manuscript text should be described method for obtaining these results. Level of protein expression must be quantified with several repeats in all Figures with Western blotting analysis of intracellular proteins expression (l means figures Fig.2A, 2E, 3C, 4D, 5D, 6D). it is not clear what the authors meant in Fig. 2F. Description of this experiment must improve. Missing Western blot verification of TTK shRNA in CL1-5 cells. Must be added. Missing microarray analysis in Materials and Methods. It is not clear how many genes changed expression by TTK-knockdown in A549 cells. No upregulated genes? Authors should explain that GAPDH is a good reference gene in their case, because sometimes GAPDH isn’t good control expression gene. In Materials and Methods must added additional details about NGS analysis.Author Response
Dear Reviewer,
We appreciate your suggestions and comments in our manuscript entitled “Title: Upregulation of Thr/Tyr kinase Increases the Cancer Progression by Neurotensin and Dihydropyrimidinase-Like 3 in Lung Cancer” (ijms-708835). We have answered the questions in a point-by-point fashion, and the manuscript has been revised based on your suggestions. All the major changes are highlighted in the text of revised one by using the “yellow highlight”.
In summary, we are grateful for the opportunity to have made these changes and clarify the concerned points. We believe that we have successfully addressed your concerns and comments. We hope the quality of our revised manuscript can fit your suggestions and comments.
Sincerely yours,
Ying-Ming Tsai M.D., on behalf of all authors
Division of Pulmonary and Critical Care Medicine, Department of Internal Medicine, College of Medicine, Kaohsiung Medical University
No. 100, Shih-Chuan 1st Road, Kaohsiung 807, Taiwan
Phone: +886-975356101
E-mail: yingming@kmu.edu.tw
Reviewer 1
In this article, Dr. Ying-Ming Tsai and colleagues presented new results about involving of serine/threonine kinase TTK in development of lung cancer. The paper is well written and potentially contains new information about TTK signaling. However, there are a major problems without correcting which this article cannot be published. I noticed the following problems:
- The Figure 1D is poorly described. In figure legend or manuscript text should be described method for obtaining these results.
Thanks for your comment, we have improved the description of IHC staining in Figure 1D. Also, we modified the description in both
Legend: (D) The IHC staining of TTK of the specimens of 6 out of 9 patients lung cancer patients, the higher intensity of TTK staining in the tumor parts when comparing with normal lung parts.
Text: To validate it, immunohistochemical (IHC) staining of TTK showed the higher expressions in lung cancer parts when comparing with normal parts (Figure 1D) from 8 (6 in Figure 1D and 2 in Figure 6E) out of 9 patients with lung cancer.
Materials and methods:
- Level of protein expression must be quantified with several repeats in all Figures with Western blotting analysis of intracellular proteins expression (I mean figures Fig.2A, 2E, 3C, 4D, 5D, 6D).
Thanks for your comments. Each of Western blotting study has been performed independently at least 3 times to verify the findings. Also, we have quantified the levels of proteins in Western blotting analysis acquired from 3 independent experiments in Figure 2A, 2E, 3C, 4D, 5D and 6D.
- It is not clear what the authors meant in Fig. 2F. Description of this experiment must improve.
We thank your constructive comments, and we have modified the description in Fig. 2F as bellow “Consistently, the cell cycle analysis revealed the S-phase population was decreased (from 19.45% to 12.37%) in TTK knockdown cancer cells, consistent with BrdU incorporation analysis (Figure 2F).”
- Missing Western blot verification of TTK shRNA in CL1-5 cells. Must be added.
We would like to thank the reviewer for careful and thorough reading of this manuscript. We missed the CL1-5 TTK knockdown data and added CL1-5 TTK knockdown data in Figure 3C. The TTK knockdown efficiency was greater than 50% which was adequate for a loss-of-function study.
- Missing microarray analysis in Materials and Methods.
Thanks for reviewer’s careful and thorough reading of this manuscript. The missing part of microarray analysis in Materials and Methods has been added in this section.
- It is not clear how many genes changed expression by TTK-knockdown in A549 cells. No upregulated genes?
As the reviewer’s concern, the data of microarray after TTK knockdown showed 19 genes change. As shown in heat map of Figure 4A, there are only 8 genes upregulation and 11 genes downregulation.
- Authors should explain that GAPDH is a good reference gene in their case, because sometimes GAPDH isn’t good control expression gene.
We agreed the comments from the reviewer that GAPDH might be changed under some stress. In fact, we used several internal control including GAPDH, actin and tubulin. We found that the results were consistent, thus we selected GAPDH as an internal control due to the molecular weight of GAPDH.
- In Materials and Methods must added additional details about NGS analysis.
Thanks for reviewer’s careful and thorough reading of this manuscript. The missing part of NGS analysis in Materials and Methods has been added in this section.

Reviewer 2 Report
The manuscript "Upregulation of Thr/Tyr kinase Increases the Cancer Progression by Neurotensin and Dihydropyrimidinase-Related Protein 3 in Lung Cancer" describes downregulation of TTK and DPYSL3 leading to reduced proliferation of lung cancer cell lines A549 and CL1-5. Data presented in the manuscript needs improvement to accept the conclusion of the authors.
Major points to consider
Authors claim that their previous publication (ref 10) has shown upregulation of TTK. However, their publication does not show TTK upregulation in the 3 pairs of normal and lung adenocarcinoma samples authors have compared. They also do not show upregulation of TTK or DPYSL3 in their comparative analysis of the various data sets. They seem to be using different databases to look at differentially expressed genes in tumor vs normal and perform in vitro studies. Survival curves again are presented for TTK or DPYSL3 from the database analyses. They need to compare normal vs lung cancer of their population (may be at least 10 pairs) to see whether the lung cancer of their population show upregulated expression of these two genes. Then, it will be valuable to investigate the effect of downregulation or upregulation genes in lung cancer cell lines. While CL1-5 was derived from an adenocarcinoma of the lung, A549 represents non small cell carcinoma of the lung. Thus, differential expression they have studied earlier from adenocarcinoma (ref 10) may be applicable to cell line CL1-5 and not A549. Western blots for CDK2 in figures 2, and 5, snail, α-SMA and vimentin in figure 3 and vimentin in figure 6 are not convincing to show decreased expression with the downregulation of TTK or DPYSL3 shRNAs. It is surprising that there are no soft agar colonies with 50% reduction of TTK in the shRNA treated A549 cells shown in figure 2B. Expression of TTK in control shRNA/A549 treated cells shown in figure 7E is not convincing for the lung metastatic regions. Tumor growth suppression of A549 cells containing TTK shRNA in nude mice is an important finding. Improvement in IHC or western blots for the control vs TTK downregulation will be valuable. Western blots need protein molecular weights as shown in figure 2A. and 4D.Author Response
Dear Reviewer,
We appreciate your suggestions and comments in our manuscript entitled “Title: Upregulation of Thr/Tyr kinase Increases the Cancer Progression by Neurotensin and Dihydropyrimidinase-Like 3 in Lung Cancer” (ijms-708835). We have answered the questions in a point-by-point fashion, and the manuscript has been revised based on your suggestions. All the major changes are highlighted in the text of revised one by using the “yellow highlight”.
In summary, we are grateful for the opportunity to have made these changes and clarify the concerned points. We believe that we have successfully addressed your concerns and comments. We hope the quality of our revised manuscript can fit your suggestions and comments.
Sincerely yours,
Ying-Ming Tsai M.D., on behalf of all authors
Division of Pulmonary and Critical Care Medicine, Department of Internal Medicine, College of Medicine, Kaohsiung Medical University
No. 100, Shih-Chuan 1st Road, Kaohsiung 807, Taiwan
Phone: +886-975356101
E-mail: yingming@kmu.edu.tw
Reviewer 2
The manuscript "Upregulation of Thr/Tyr kinase Increases the Cancer Progression by Neurotensin and Dihydropyrimidinase-Related Protein 3 in Lung Cancer" describes downregulation of TTK and DPYSL3 leading to reduced proliferation of lung cancer cell lines A549 and CL1-5. Data presented in the manuscript needs improvement to accept the conclusion of the authors.
Major points to consider
- Authors claim that their previous publication (ref 10) has shown upregulation of TTK. However, their publication does not show TTK upregulation in the 3 pairs of normal and lung adenocarcinoma samples authors have compared. They also do not show upregulation of TTK or DPYSL3 in their comparative analysis of the various data sets.
Thanks for reviewer’s careful and thorough reading of this manuscript. The previous 2 pairs of normal and lung adenocarcinoma tissues showed higher expression levels of TTK out of 3 patients. One of these three did not reach the threshold of 2-fold change. However, we collected another specimens from lung cancer patients which showed higher expression levels of TTK. So, the potential of TTK mediating lung cancer development was high. And we chose it as our target.
Most importantly, we also provided the sequential sections for TTK/DPYSL3 using IHC staining. The result showed TTK/DPYSL3 were positively correlated in lung cancer tissues. The result was shown on Figure 6E.
- They seem to be using different databases to look at differentially expressed genes in tumor vs normal and perform in vitro studies. Survival curves again are presented for TTK or DPYSL3 from the database analyses.
Thanks for your suggestions. We have re-analyzed the dataset of Okayama (GSE 31210, a public cohort to provide information) to analyze the survival curves of TTK and DPYSL3. Whenever possible, we kept the same databases, GSE 31210, for survival analysis. In Figure 1E and 6F, we have added the survival analysis of TTK and DPYSL3 from GSE 31210. There were shorter survival time when TTK or DPYSL3 expresses higher levels.
- They need to compare normal vs lung cancer of their population (may be at least 10 pairs) to see whether the lung cancer of their population show upregulated expression of these two genes.
As suggested, we have analyzed the protein expression of TTK and DPYSL3 in 8 persons via IHC staining and added in in Figure 1D (6) and Figure 6E (2). The sequential section of tissue for IHC staining in both TTK and DPYSL3 and the result showed positive correlation in their protein expressions (Figure 6E).
- Then, it will be valuable to investigate the effect of downregulation or upregulation genes in lung cancer cell lines. While CL1-5 was derived from an adenocarcinoma of the lung, A549 represents non-small cell carcinoma of the lung. Thus, differential expression they have studied earlier from adenocarcinoma (ref 10) may be applicable to cell line CL1-5 and not A549.
We appreciated your comment. For this part of study, we performed the migration of this study using CL1-5. For CL1-5 was aimed to study of cancer cell migration.
- Western blots for CDK2 in figures 2, and 5, snail, α-SMA and vimentin in figure 3 and vimentin in figure 6 are not convincing to show decreased expression with the downregulation of TTK or DPYSL3 shRNAs.
As suggested, we changed the images to a higher quality ones and also quantified the results of Western blotting. Other than improved quality of image, we also measured the density of each band in three independent study which showed the difference of CDK2 in Figure 2 & 5, snail, α-SMA and vimentin in Figure 2 and vimentin in Figure 6.
- It is surprising that there are no soft agar colonies with 50% reduction of TTK in the shRNA treated A549 cells shown in figure 2B.
Thanks for your comment. The colony formation assay but not soft agar was performed in Figure 2B. The Figure 2B represented one of triplicate experiment. Also, we provided another set of the triplicate.
- Expression of TTK in control shRNA/A549 treated cells shown in figure 7E is not convincing for the lung metastatic regions. Tumor growth suppression of A549 cells containing TTK shRNA in nude mice is an important finding. Improvement in IHC or western blots for the control vs TTK downregulation will be valuable.
As suggested, we have improved the quality of image which revealed the difference of TTK expression after shRNA knockdown. The Figure 7E showed lower expression level of TTK after shRNA knockdown.
- Western blots need protein molecular weights as shown in figure 2A. and 4D.
As suggested, we have labeled the molecular weights in Figure 2A and 4D.
Round 2
Reviewer 1 Report
The manuscript has been improved, and my comments have been addressed. I recommend to accept this article for publication.
Author Response
Dear Reviewer,
We appreciate your suggestions and comments in our manuscript entitled “Title: Upregulation of Thr/Tyr kinase Increases the Cancer Progression by Neurotensin and Dihydropyrimidinase-Like 3 in Lung Cancer” (ijms-708835). We have answered the questions in a point-by-point fashion, and the manuscript has been revised based on your suggestions. All the major changes are highlighted in the text of revised one by using the “yellow highlight”.
In summary, we are grateful for the opportunity to have made these changes and clarify the concerned points. We believe that we have successfully addressed your concerns and comments. We hope the quality of our revised manuscript can fit your suggestions and comments.
Sincerely yours,
Ying-Ming Tsai M.D., on behalf of all authors
Division of Pulmonary and Critical Care Medicine, Department of Internal Medicine, College of Medicine, Kaohsiung Medical University
No. 100, Shih-Chuan 1st Road, Kaohsiung 807, Taiwan
Phone: +886-975356101
E-mail: yingming@kmu.edu.tw
Review 1
Comments and Suggestions for Authors
The manuscript has been improved, and my comments have been addressed. I recommend to accept this article for publication.
Thanks for your comment, we will keep on further advanced study in this field.
Reviewer 2 Report
The revised version of the manuscript is improved. However, there are errors that need to be addressed
- M.wt of TTK is marked near a marker of 95kD in figure 3C, however it is marked near a marker of 96kD. Which of the markers is correct
- M.wt of DPYSL3 is shown near the marker 62kD in figure 4D which is expected for this protein. TTK shRNA reduces the expression indicating the regulation of this protein by TTK. In figure 6D, loss of a 95kD protein is shown for DPYSL3 with the siRNA. loss of endogenous expression should point to the loss of 62kD and there is no loss for the protein at 62kD. Either there is an error in m.wt markers or a different protein (related or a variant) is affected by DPYSL3 siRNA. This needs explanation and/or correction.
- line 269 shows "inhibition of TTK leads to increased expression of NTS". It needs to be corrected to "inhibition of TTK leads to decreased expression of NTS"
Author Response
Dear Reviewer,
We appreciate your suggestions and comments in our manuscript entitled “Title: Upregulation of Thr/Tyr kinase Increases the Cancer Progression by Neurotensin and Dihydropyrimidinase-Like 3 in Lung Cancer” (ijms-708835). We have answered the questions in a point-by-point fashion, and the manuscript has been revised based on your suggestions. All the major changes are highlighted in the text of revised one by using the “yellow highlight”.
In summary, we are grateful for the opportunity to have made these changes and clarify the concerned points. We believe that we have successfully addressed your concerns and comments. We hope the quality of our revised manuscript can fit your suggestions and comments.
Sincerely yours,
Ying-Ming Tsai M.D., on behalf of all authors
Division of Pulmonary and Critical Care Medicine, Department of Internal Medicine, College of Medicine, Kaohsiung Medical University
No. 100, Shih-Chuan 1st Road, Kaohsiung 807, Taiwan
Phone: +886-975356101
E-mail: yingming@kmu.edu.tw
Review 2
Comments and Suggestions for Authors
The revised version of the manuscript is improved. However, there are errors that need to be addressed
- wt of TTK is marked near a marker of 95kD in figure 3C, however it is marked near a marker of 96kD. Which of the markers is correct
We would like to thank the reviewer’s careful and thorough reading of this manuscript. The molecular weight of TTK is 95kDa and we wrote the M. wt erroneously and have modified it into correct one (Figure 2A and 4D).
- wt of DPYSL3 is shown near the marker 62kD in figure 4D which is expected for this protein. TTK shRNA reduces the expression indicating the regulation of this protein by TTK. In figure 6D, loss of a 95kD protein is shown for DPYSL3 with the siRNA. loss of endogenous expression should point to the loss of 62kD and there is no loss for the protein at 62kD. Either there is an error in m.wt markers or a different protein (related or a variant) is affected by DPYSL3 siRNA. This needs explanation and/or correction.
Thanks for reviewer’s careful and thorough reading of this manuscript. The correct M. wt of DPYSL3 is 62kDa. We have corrected the figure into a suitable one (Figure 4D and 6D) which indicated the M. wt of DPYSL3 as 62kDa with suppressed expression after siRNA by Western blotting (Figure 6D).
- Line 289 shows "inhibition of TTK leads to increased expression of NTS". It needs to be corrected to "inhibition of TTK leads to decreased expression of NTS"
Thanks for reviewer’s thorough and detailed reading of this manuscript for our error in this point. .We have corrected it into the following description (line 289) as “In this study, we found that inhibition of TTK decreased the expression of NTS, which was observed by microarray, but also are validated by protein expression analysis. Furthermore, addition of exogenous NTS restore cell proliferation, cyclin A and cdk2 expression, which were inhibited by TTK deficiency.”